# Monte Carlo Thought Search: Large Language Model Querying for Complex Scientific Reasoning in Catalyst Design

**Henry W. Sprueill[1], Carl Edwards[2], Mariefel V. Olarte[1], Udishnu Sanyal[1],**

**Heng Ji[2], Sutanay Choudhury[1]**

[1]Pacific Northwest National Laboratory, Richland, Washington, USA
[2]University of Illinois Urbana-Champaign, Urbana, Illinois, USA
{henry.sprueill, mariefel.olarte, udishnu.sanyal, sutanay.choudhury}@pnnl.gov
{cne2, hengji}@illinois.edu

## Abstract

Discovering novel catalysts requires complex reasoning involving multiple chemical properties and resultant trade-offs, leading to a combinatorial growth in the search space. While large language models (LLM) have demonstrated novel capabilities for chemistry through complex instruction following capabilities and high quality reasoning, a goal-driven combinatorial search using LLMs has not been explored in detail. In this work, we present a Monte Carlo Tree Search-based approach that improves beyond state-of-the-art chain-of-thought prompting variants to augment scientific reasoning. We introduce two new reasoning datasets: 1) a curation of computational chemistry simulations, and 2) diverse questions written by catalysis researchers for reasoning about novel chemical conversion processes. We improve over the best baseline by 25.8% and find that our approach can augment scientist's reasoning and discovery process with novel insights.[1]

## 1 Introduction

Scientific discovery thrives on uncovering the optimal combinations of factors that maximize a property of interest. For example, to discover new efficient fuels (Yang et al., 2019; Tran et al., 2023; Zitnick et al., 2020) or chemical conversion processes requiring less energy, a scientist would need to consider the chemical reaction, the reactants that undergo the reaction, the catalysts that improve the rate of reaction, and find the optimal combination of operating conditions (Fig. 2). Mathematically, one could represent this as an optimization problem where we model a chemical process as a function and formulate the search problem as finding the optimal combination of all process parameters that minimizes a cost function modeled around energy efficiency. For highly empirical fields such as chemistry, these combinatorial searches require

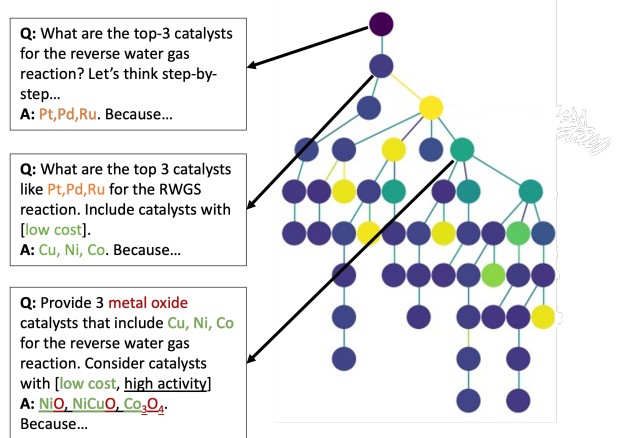

Figure 1: An example prompt design via tree search. The search begins with a generic query at the root node. The answer from each node is passed to the child nodes and additional criterion are added to the prompt. For instance, low cost. Information passed to children nodes is color coded to show the reasoning pathway.

expert reasoning with knowledge of the scientific literature that dates back a century. The emerging capability of large language models (LLMs) (Wei et al., 2022; Ouyang et al., 2022; Taylor et al., 2022; Lai et al., 2023; OpenAI, 2023) provides an opportunity to automatically reason with a large knowledge space in a human-interpretable way.

Despite their promise, the brittleness of language models to their inputs and hallucination remain areas for concern (Creswell and Shanahan, 2022; Taylor et al., 2022). Our initial investigation of LLMs revealed that basic prompting (such as "What is a good catalyst for reaction X?") leads to basic answers that could be found on a Wikipedia page. To improve the quality of answers, one can incorporate desirable properties into the prompt which lead the LLM to produce more specific answers (such as "What is a good catalyst with low cost for reaction X?"). Additionally, LLMs often hallucinate, producing answers without grounding in scientific fact. Achieving accurate answers with high speci-

---

[1]Our codebase and datasets are freely available at https://github.com/pnnl/chemreasoner

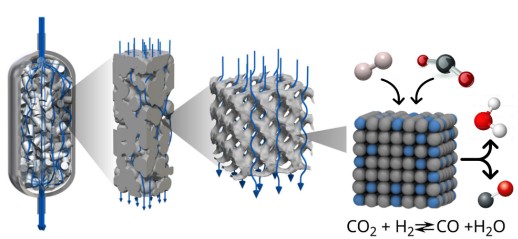

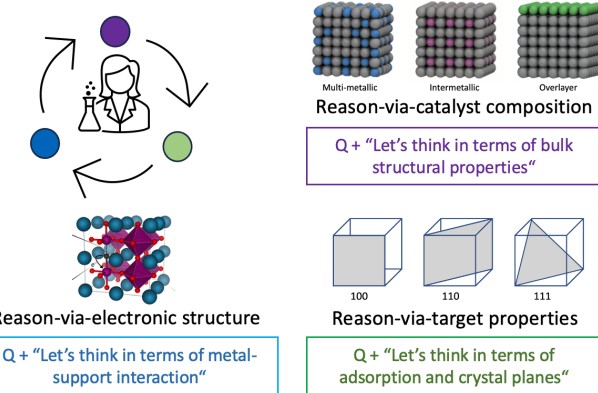

**Reason-via-catalyst composition**

Q + "Let's think in terms of bulk structural properties"

**Query:** What are the top-3 catalysts that are cheap, perform reverse water gas shift reaction at lower temperature (<200 C) and demonstrate higher adsorption energy for both CO₂ and H₂?

Reason-via-electronic structure

Q + "Let's think in terms of metal-support interaction"

**Reason-via-target properties**

Q + "Let's think in terms of adsorption and crystal planes"

$CO_2 + H_2 \rightleftarrows CO + H_2O$

Figure 2: Illustration of the combinatorial thinking used by human experts to reason about a catalyst (best viewed in color). They successively "think in terms of" different constraints and factors, each of which are related via scientific principles, and narrow down the set of possible candidates. Our Monte Carlo Reasoner emulates such cognitive thinking by prompting a language model with different combinations, yielding a tree-structured space of queries and potential candidates, and returns the optimal answer via efficient exploration of the possible space.

ficity and which use key technical terminology (Fig. 2) is essential to earn the scientific community's trust and pave the way for the adoption of machine reasoning systems.

In this work, we focus on the problem of prompting an LLM to return the top-$k$ catalysts for a given chemical reaction and generating the reasoning for each candidate. In collaboration with researchers from the catalysis community, we develop a new dataset, BioFuels Question Reasoning (BioFuelQR), consisting of complex reasoning questions and answers. We observe the reasoning pathways used by domain scientists and conclude that it is important for LLMs to progress from "thinking step-by-step" to "thinking step-by-step in terms of relevant properties". In this setting, we are given a question which has some relevant catalyst properties $\mathbf{P}, |\mathbf{P}| = n$ (e.g. {"crystal planes", "toxicity"}) and we want to identify the best subset $\mathbf{R} \subset \mathbf{P}, |\mathbf{R}| = r$ of properties for the language model to "think" in terms of. Considering that language models are sensitive to permutations of their inputs, there are $P_r^n = \frac{n!}{(n-r)!}$ possible prompts to search through. This goal can be accomplished by learning to prompt the LLM with the most relevant subset of properties (Deng et al., 2022) or decomposing the set into a sequence of chained queries (Dohan et al., 2022). In both cases, identification of the prompt-generating property subset becomes the limiting factor.

To solve this problem, we propose the M̲onte C̲arlo R̲easoner (MCR), a generic heuristic search methodology that addresses the combinatorial chal-

lenge of query decomposition. Considering the practical challenges of learning prompts that are both human comprehensible (a key consideration for scientists) and provide the best performance (Deng et al., 2022), we pursue a stochastic, heuristic search-based approach that leverages LLMs trained on scientific literature with sophisticated instruction following capabilities (Ouyang et al., 2022).

We formulate the task as a search problem in which an agent performs a query in an uncertain environment (represented by the LLM) and determines a query variant to pursue based on the evaluated reward. Given an initial query, we construct a tree structure of these unique query variants in order to progressively refine the original query (the root) into property-specific variations (the leaves). Our methodology demonstrates improvement over basic querying of the LLM without any additional training of the LLM. Instead, we use a Monte Carlo Tree Search algorithm (MCTS) to perform a stochastic search over the existing knowledge space of an LLM to achieve more scientifically valuable answers.

Our second major contribution is demonstrating the efficacy of using a scientific domain-specific reward function in LLM-based computations for our top-$k$ catalyst problem. Estimation of the "adsorption energy" of a chemical structure is at the core of developing efficient chemical reactions (see Appendix A.2 for details). Finding catalysts that can enable chemical reactions with the least amount of external energy is key to developing environmen-

tally friendly industrial processes. In this work, we implement such energy function specific considerations via a LLM-derived reward function. Our experiments (using questions detailed in Table 4) show that even a simplistic reward function dramatically improves the specificity of answers and their associated reasoning from the LLM.

In summary, we make the following contributions:

1. We present *Monte Carlo Reasoner (MCR)*, an algorithm to prompt LLMs for zero-shot complex reasoning tasks involving combinatorial search.
2. We introduce a new chemistry-focused dataset, BioFuelQR, that captures key reasoning challenges in hypothesis generation and testing faced daily by scientists. We present in-depth qualitative analysis of MCR on BioFuelQR.
3. We demonstrate that a domain-specific reward function that represents a fundamental scientific concept can lead to dramatic improvement in the quality and specificity of LLM answers.

## 2 Monte Carlo Reasoner

**Problem definition**    Our goal is find the optimal prompt, $P^o$, which leads the LLM to return the best candidate catalysts for a specific problem. Starting with a general initial prompt $P_0$, we use a set of actions to automatically modify the prompt to improve the LLM output with respect to a reward function, R.

For instance, suppose $P_0$ is the prompt given in Figure 2(left). Each prompt is a template, where we use actions $a \in \mathcal{A}$ to create better prompts, based on how experts might modify their own queries, so that the LLM will suggest superior catalysts. See Appendix C.1 for a more detailed explanation of the actions and prompt. By modifying prompts, we create a tree of prompts, answers, and rewards, as demonstrated in Figure 1. We call a path from the root to a leaf node a "reasoning pathway". These reasoning pathways can be constructed in several different ways. For instance, we can take an action to introduce additional catalyst properties to consider (such as "composition of metals" and "electronic structure" in Fig. 2 (right)) so that the LLM will include or exclude certain catalysts in its answer. Also, for each prompt $P$ after $P_0$, we include $P$'s parent node's answer in $P$ to provide the LLM with additional context about the previous answer. Further, at each node, we prompt the LLM to produce catalysts with either "new elements", "similar elements", or "different elements"

to the parent node's answer candidates (switching between these possibilities is an action). Finally, we can take an action to change the type of catalyst requested (unary, binary, ternary, or -oxide catalysts). Clearly, the number of possible reasoning pathways increases drastically with tree depth due to the possible combinations of actions. Thus, we apply Monte Carlo Tree Search, an efficient method to optimize a sequence of actions with a reward function, $R$.

In MCTS, each prompt $P$ is stored as a node in a tree $T$, where edges are prompt-action pairs $(P_i, a_j)$. The search tree decides at each prompt which action to take to obtain the best reward based on previous results. Typically, prompt-action pairs are weighted by a policy, which determines *a-priori* the importance of each action for a prompt, given as prior probabilities. Here, we assign equal weight to all possible actions. Impossible actions are assigned weight of 0 (see Appendix C.2).

In MCTS, each edge stores a count $N(P,a)$, a weight representing a prior probability $p(P,a)$, and the total downstream reward $V(P,a)$ where

$$V(P,a) = \sum_{P' \in \text{successor}(P)} \gamma^d R(P') \quad (1)$$

Here, $\gamma$ is a discount factor and $d$ is the (tree) distance of $P'$ from $P$. If there are no discovered successors to $P$, then we set $V(P,a) = 0$. The search determines the next action to take with policy $\mathcal{P}(V, N, p)$:

$$\underset{a_i \in \mathcal{A}}{\text{argmax}} \left( \frac{V(P,a_i)}{N(P,a_i)} + cp(P,a_i) \frac{\sqrt{\sum_j N(P,a_j)}}{1+N(P,a_i)} \right) \quad (2)$$

where $c$ is an exploration-exploitation trade-off. The simulation starts at the root node each time and traverses the constructed tree until a new state is reached. Then, its answer and reward are calculated, stored, and the upstream values of $V$, $N$ are updated. This is repeated to generate the desired number of prompts (in our case 300). MCTS is superior to re-sampling methods because it avoids repeatedly sampling the same prompt and it is superior to brute-force tree search methods such as BFS and DFS because it selects trajectories in the tree that show promising results.

### 2.1 Reward Function

Our reward function, $R$, measures the effectiveness of the catalysts proposed by the LLM for a given prompt, $P$. Here, we measure effectiveness

**Algorithm 1:** Run MCR search. Here, $a^t$ indicates $t^{\text{th}}$ action from root.

---

1  **Require:** LLM, initial prompt $P_0$, number of candidate catalysts $k$, number of prompts to generate $M$
2  Initialize tree $T$. Define nodes $P$ and edges $(P, a_j)$, discount $\gamma$, stored values $N(P, a_j)$, $V(P, a_j)$, $p(P, a_j)$, and reward function $R$.
3  $root(T) \leftarrow P_0$
4  **for** $j = 1, \ldots, M$ **do**
5  $\quad$ Current Node $P_0 = root(T)$
6  $\quad$ Current Depth $t \leftarrow 0$
7  $\quad$ **while** $P_t \in T$ **do**
8  $\quad\quad$ Select $a^t \sim \mathcal{P}(P_t, a_i, N, V, p)$
9  $\quad\quad$ Increment $N(P_t, a^t)$
10 $\quad\quad$ $P_{t+1} = a^t(P_t)$ $\qquad \triangleright$ Apply action
11 $\quad\quad$ Increment $t$
12 $\quad$ **end**
13 $\quad$ Send $P_t$ to LLM
14 $\quad$ Save $P_t$, $(P_t, a^t)$ in $T$
15 $\quad$ $r \leftarrow R(P_t)$ $\qquad \triangleright$ Calculate reward using answer from LLM
16 $\quad$ **for** $t' = t - 1, \ldots, 0$ **do**
17 $\quad\quad$ $V(P_t', a^{t'}) = V(P_t', a^{t'}) + \gamma^{t-t'} r$
18 $\quad$ **end**
19 **end**
20 **return** $\arg\max_{P \in T}(R(P))$

---

Table 1: Final catalyst suggestion results. $N_P$ is number of prompts evaluated and $d_{max}$ is maximum search tree depth. Values are averaged over evaluated examples.

| Method | OpenCatalysis | | | BioFuelQR | | |
|---|---|---|---|---|---|---|
| | Reward | $N_P$ | $d_{max}$ | Reward | $N_P$ | $d_{max}$ |
| CoT | 2.04 | 1 | N/A | 2.27 | 1 | N/A |
| CoT w/ Self-consistency | 4.04 | 10 | N/A | 6.38 | 10 | N/A |
| ToT (breadth-first-search) | 9.91 | 253 | 5 | 13.8 | 253 | 5 |
| MCR (ours) | 12.47 | 301 | 9.33 | 15.6 | 301 | 9.5 |

for an example). We collected two answers from catalysis researchers for a subset of 51 queries to observe different reasoning patterns and human biases. See section C for details on dataset design.

**Baselines** We benchmark MCR's performance with three recent methods: 1) Chain-Of-Thought (CoT) prompting (Kojima et al., 2022), 2) Self-consistency-based CoT (Wang et al., 2022), 3) breadth-first-search (BFS) based Tree-of-Thoughts (ToT) (Yao et al., 2023) (a contemporary work to ours). Experiments are based on GPT-3 text-davinci-003 [2]. Table 1 shows MCR improves by 25.8% and 13% over the reward obtained by BFS on OpenCatalysis and BioFuelQR, respectively. Performance improves by $\sim$600% over CoT.

**Query Cost** Despite significant effort with the dataset creation, our results in Table 2 are obtained from 11/130 queries. MCR and baselines are implemented using OpenAI text-davinci-003 for consistency. MCR and ToT method is computationally expensive (Table 2), so evaluation of all 130 queries over all methods requires approximately 174,470 API calls, and we could not secure compute capacity from OpenAI to evaluate more than 11 queries with each method. We further discuss the limitations that arose in Limitations (4).

**Key Takeaways** We find that MCR's use of stochastic search prunes the more uniform exploration of search space conducted by ToT (Yao et al., 2023). Table 2 shows given a maximum query limit, MCR was able to search significantly deeper (reported by $d_{max}$) than ToT. While MCR reached a higher reward than ToT, MCR generated more nodes than ToT (see C.4). However, we are not able to definitively declare that both tree-based methods outperformed CoT and CoT w/ Self-consistency.

To confirm if the increased reward over CoT indeed translates into better reasoning quality, two catalysis experts compared the best answer generated by MCR with the GPT-3 CoT implementation. Overall, the experts preferred MCR to CoT (Fig-

of a catalyst by querying the LLM to produce adsorption energies for a given adsorbate in electron volts (eV). We describe the prompt used to generate the adsorption energy in Appendix C.1. The significance of adsorption energy for catalysis design is explained in Appendix A.2. The reward is calculated as

$$R(P) = \sum_{a \in \text{ adsorbates}} |LLM(a, C(P))| \quad (3)$$

where $C(P)$ is the top-$k$ catalysts from prompt $P$.

## 3 Experiments

**Experimental setup** We conduct our experiments on two new chemistry-focused reasoning query benchmarks containing 130 queries (Table 2). We compile OpenCatalysis from the OC20 (Chanussot et al., 2010) and OC22 (Tran et al., 2023) catalyst datasets (Zitnick et al., 2020). Second, we develop BioFuelQR–a query dataset targeting biofuels-focused catalyst discovery (see Table 4

---

[2]OpenAI has now discontinued support for this model. In the future, using open models to avoid this issue would be desirable, but we found their performance lacking.

Figure 3: Domain expert evaluation of LLM answers on the reasoning path to the final node with highest reward.

ures 8, 9). Figures 5-9 illustrates the evaluation for one such query. Review from both experts (Figures 8, 9) deemed the CoT response in Figure 6 incorrect and MCR correct (Figure 8).

The experts also evaluated how the prompts and LLM answers evolve as MCR searches deeper in the prompt tree (Figures 3 and 10)–in many cases they found the LLM answers to be logically coherent and in some cases even insightful enough for follow-up experimentation (see the second user feedback in Figure 3). Overall, both experts preferred MCR for having higher specificity over CoT and reasoning in terms of correct properties (detailed in Figures 8, 9).

## 4    Conclusion and Future Work

LLMs offer major promise to automate the cycle of scientific hypothesis generation and testing. Our work tackles the challenge of identifying key prop-erties for augmenting a chemist's reasoning via use of a domain-specific reward function, enabling generation of relevant scientific explanations with high specificity. MCR is a zero-shot reasoning methodology that circumvents the need for large-scale, hard-to-obtain, domain-specific training data. We apply it to catalyst research: a highly empirical, reasoning-heavy scientific field dating back a century. Future work can investigate large-scale evaluation of our benchmark, integration with atomistic prediction models trained on quantum chemistry datasets for more trustworthy reward functions, and finetuned language models.

## Limitations

In this work, we consider applications of large language models in the scientific domain. In general, this comes with a number of limitations. First, LLMs display largely black box behavior, which is exacerbated by many strong models only being accessible as APIs. Second, generative modeling in the scientific domain is incredibly difficult to evaluate, requiring laboratory verification in many settings. Third, hallucination about factual information is a concern. One benefit of our method is that it provides reasonings based on refined prompts, which we show can be inspirational to domain experts searching for a solution.

Our work demonstrates that tree-search methods have a strong value proposition over existing methods for LLM reasoning (CoT, self-consistency etc.). Since ToT is contemporary to our methodology, an important contribution of this work is demonstrating the merit of tree-based reasoning approaches for complex scientific reasoning tasks; scientific reasoning is not discussed in (Yao et al., 2023). We do not claim that MCR is necessarily superior to ToT in all settings. In fact, further experiments have shown the two methods can be quite comparable. However, we are limited in this work by the cost of experimentation that we cannot perform an ideal comparison of MCR to ToT.

In particular, our reward function based on LLM outputs of scientific questions can be considered a limitation. However, it allows for much quicker validation of ideas and we find it to be an effective proxy (which on its own is interesting). In the future, comparatively costly atomistic simulations can be used to replace our reward function. These can be quite time-consuming and computationally expensive, so we focus on our algorithmic

contribution in this work. Because of the efficacy we demonstrate using LLM rewards, it may also be possible to use a hybrid approach to save on computational chemistry simulations. This could initially leverage LLM embeddings as an initial reward to narrow down promising search sub-trees by selecting the most promising nodes in the first few layers of the search tree. Advanced simulations can then be used for searching final answers in these sub-trees. Alternatively, simulations can be used as a limited-use oracle like in active learning. We leave this for future work.

Our method's improvement comes with higher cost of inference, similar to Tree-of-Thought. When doing inference locally, this may not be a problem. However, we utilize third-party APIs which are both expensive and rate-limited. We found existing open-source models trained on chemistry text did not possess sufficient instruction-following capabilities to be reliable or effective here. Thus, we were limited in quantity of experiments that could be done, as well as the models which could be accessed. This is because our approach requires an average of 750 API calls per tree search. Although we evaluate on relatively few initial questions, our in-depth expert-performed analysis is based on $\sim$7,200 queries.

## Ethical Considerations

We propose a zero-shot prompting methodology for LLMs that enables reasoning for complex queries in the scientific domain. Like most applications of LLMs, this has similar ethical considerations, especially in regards to implicit biases from large-scale pretraining and the hallucination of false information. Thus, it is still important for human oversight and careful evaluation of language model output. One consideration of our method is that it may enable discovery of molecules, materials, or other scientific products which can be used for harmful applications (Urbina et al., 2022). Overall, we believe these downsides are outweighed by the benefits of this work to both the NLP community and other scientific communities which may benefit.

## Acknowledgements

This work is supported in part by the PNNL seed Laboratory Directed Research and Development (LDRD) program, Data-Model Convergence initiative. This work is also in part based upon work supported by the Molecule Maker Lab Institute: an AI research institute program supported by NSF under award No. 2019897 and No. 2034562. The views and conclusions contained herein are those of the authors and should not be interpreted as necessarily representing the official policies, either expressed or implied, of the U.S. Government. The U.S. Government is authorized to reproduce and distribute reprints for governmental purposes notwithstanding any copyright annotation therein.

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

# A Background

## A.1 Scientific Drivers from Catalysis

Discovery of novel catalysts is essential for accelerating the transition to a sustainable future. Despite the significant progress in the development of highly efficient catalysts, heterogeneous catalysis remains largely an empirical science owing to the complexity of the underlying surface chemistry (Nørskov et al., 2011). Currently, there is a lack of data and design guidelines for heterogeneous catalysis because the computational cost of obtaining accurate theoretical models for such complex systems is currently prohibitively high while high-throughput experimental methods that have been applied successfully in related fields have not yet been thoroughly explored (Yang et al., 2019).

Experimental validation of a new catalyst and its performance is expensive (Yang et al., 2019). Artificial intelligence-driven computing approaches aims to accelerate such discovery by down-selecting candidates that are most promising and merit extensive evaluation in a laboratory (Ward et al., 2021). The past few years have seen a lot of developments for applying AI to chemistry that range from predicting properties of atomistic structures, or outcomes of reactions (Schwaller et al., 2019; Chen and Jung, 2022). Generative models (Jin et al., 2018) or deep reinforcement learning methods (You et al., 2018) have demonstrated abilities to propose novel chemical compounds that satisfy unique property constraints, and then suggest synthesis pathways for producing such compounds (Struble et al., 2020). Generally, such models are trained on representations of atomistic structures, or reactions between multiple structures (Struble et al., 2020; Chen and Jung, 2022).

## A.2 Motivation for molecular energy prediction as a reward function

Electronic structure calculations play a crucial role in developing atomistic-level understanding of the interaction of liquid or gaseous molecules with solids, as a functional of the topological property of the solid surface (Nørskov et al., 2011). Much of the literature from machine-learning for atomistic systems have focused on training system-level properties such as potential energy functions (Schütt et al., 2018; Gasteiger et al., 2021). The following paragraph explains why estimating the energy functions associated with a molecular structure is critical to discovering processes with lower energy requirements.

The amount of usable energy for a physical system with constant temperature and pressure is referred to as the Gibbs free energy, or Gibbs energy and is defined as: $G = H - TS$, where $H$ is the energy contained in the bonds between atoms, $T$ is the temperature and $S$ is the entropy (Zitnick et al., 2020). The entropy of a system increases when

molecules break their bonds and decreases when they form new ones. The computation of $H$ involves the potential energy between atoms. When Gibbs energy is negative, it means that the energy contained in the bonds is higher, and a system will naturally approach a lower energy state. Thus, a reaction or process will proceed spontaneously. On the contrary, a positive Gibbs energy indicates that the extrinsic energy is required to enable a target process or reaction. The path to decarbonization lies with discovering chemical processes that require lesser amount of extrinsic energy.

## B    Related work

We begin with providing an overview of the broader literature around language models and their applications into chemistry, then specifically focus on large-language models. Finally, we finish with an overview of various chain-of-thought prompting methods that have been instrumental in improving the reasoning capability of LLMs.

### B.1    Multi-modal models for Chemistry

Recently, advances in NLP have found surprising, strong results in the chemistry domain by training LLMs (Fabian et al., 2020; Chithrananda et al., 2020; Vaucher et al., 2021; Schwaller et al., 2021; NVIDIA Corporation, 2022; Tysinger et al., 2023) on string representations of molecules (Weininger, 1988; Weininger et al., 1989; Krenn et al., 2020; Cheng et al., 2023). To enable higher-level control over molecular design, multi-modal models (Edwards et al., 2021; Vall et al., 2021; Zeng et al., 2022; Xu and Wang, 2022; Su et al., 2022; Seidl et al., 2023; Xu et al., 2023; Zhao et al., 2023; Liu et al., 2023b) have been proposed. Existing work focuses on cross-modal retrieval (Edwards et al., 2021; Zeng et al., 2022), translation (Edwards et al., 2022; Liu et al., 2023c; Christofidellis et al., 2023), and editing (Liu et al., 2022).

### B.2    LLMs for Chemistry

Due to recent progress in chat-oriented models such as GPT-4 (OpenAI, 2023), interest has grown in uncovering chemical knowledge and molecular discovery from existing general LLMs (Hocky and White, 2022; White et al., 2022, 2023; Castro Nascimento and Pimentel, 2023). This has been extended to work in the few-shot setting (Ramos et al., 2023; Jablonka et al., 2023). In particular, there is an interest in endowing LLMs with scientific tools (Bran et al., 2023; Boiko et al., 2023; Liu et al., 2023a). In general, these studies assess the inherent chemistry knowledge in LLMs and the effect of integrating chemistry data via in-context learning or finetuning. This differs from our contribution, where we propose an algorithmic approach for improving model output using domain-specific rewards. A future research direction may be able to incorporate these two approaches together for exciting results.

### B.3    Chain-of-Thought (CoT) Variants

Several works have considered improving LLM output on complex reasoning tasks via formulating multiple queries. (Creswell et al., 2022) explored the decomposition of complex queries into smaller, more reliable operators. (Creswell and Shanahan, 2022) presents a methodology for generating the answer in a step-by-step fashion and uses another model or function to pick the top-ranked answers, and avoids hallucination by constraining the output to a narrower set. (Jung et al., 2022) proposed an alternate approach to generate a tree of possible explanations (both correct and incorrect), and then analyzes their relationships to infer the correct set of answers. (Wang et al., 2022) improves reliability by sampling multiple explanations and answers from the model and then selecting the final answer that appears most often. Tree-of-Thoughts (ToT) (Yao et al., 2023) generalizes the CoT approach to enable exploration over coherent units of text (thoughts) to perform deliberate decision making by considering multiple different reasoning paths. We benchmark against (Kojima et al., 2022; Wang et al., 2022; Yao et al., 2023) in our work.

## C    Dataset Design

We propose two task datasets related to catalyst design: the first is derived from the Open Catalyst (OC) Project (Zitnick et al., 2020) and the second consists of complex reasoning queries designed by catalysis experts. Our multi-disciplinary team involves researchers who actively work on designing new catalysts for bio-fuels development.

### C.1    Action-Driven Prompt Design

To apply MCR to catalyst discovery, we define a set of prompt templates and a set of actions to modify the fields of those templates. The exact structure of the prompt templates varies between

task datasets, but there are several common elements. Table 3 lists the action types that we use.

Firstly, all prompts query the language model to return "top-$k$" catalysts as , where $k$ is given by the user. Secondly, each template has a list of "include properties" and "exclude properties", which specify contexts for the LLM to consider positively when determining catalysts to include and exclude, respectively. Next, each prompt in both ToT (Yao et al., 2023) breadth-first-search and MCR after the initial prompt uses the previous list of candidate catalysts. The LLM is prompted to either include elements "similar to" or "different from" the previous list or to "include elements from" or introduce "new elements to" the list. Finally, the template includes a field to prompt for a certain kind of catalyst: unary, binary, trinary, and oxides. Of course, a prompt can have no specification on the catalyst type.

The specific template depends on the task dataset and the original query.

## C.2 Open Catalyst Dataset

The Open Catalyst project (Zitnick et al., 2020) is an online repository of datasets intended for training surrogate models for computational chemistry simulations related to catalysis. The dataset contains hundreds of thousands of adsorption energies for adsorbate-catalyst pairs calculated using density function theory (DFT), an accurate method for computing energies of atomic configurations. We use the Open Catalyst dataset to build an evaluation dataset consisting of 79 adsorbates. This dataset targets the LLM's ability to reason about the adsorption of specific adsorbates.

We use the following template for this dataset:

`Generate a list of candidate {catalyst label} {candidate list statement} for the adsorption of {adsorbate}. {include statement} {exclude statement} Let's think step-by-step and return a list of top {k} answers and their explanations as a list of pairs.`

Here, {} denote fields that need to be filled. The fields provided in the base query are the number of candidate catalysts '$k$' ($k$=5 for the OC dataset) and enters the adsorbate symbols 'adsorbate' from the OC dataset. 'Include statement' and 'exclude statement' are phrases built from the list of properties to include and exclude, respectively. These statements are affected by the Add

Table 2: Dataset Summary

|  | **OpenCatalysis** | **BioFuelQR** |
|---|---|---|
| #Queries | 79 | 51 |
| Adsorbates | ✓ | ✓ |
| Reactions | ✗ | ✓ |
| Human Answers | ✗ | ✓ |

Include Property and Add Exclude Property action types in Table 3. The 'catalyst label' field determines which kind of catalyst the LLM should return. It's value is set by the Change Catalyst Type action and the Toggle Oxide action can set this field to query for oxide catalysts. Finally, the candidate list statement is a phrase built from the list of candidates generated by the parent prompt. Since the candidate list can have an impact on the output of the LLM, we include an action to re-run the previous query with the candidate list from the previous query's output.

Possible actions are weighed with equal prior probabilities $p$ (see Section 2) and impossible actions are given prior probability zero. Actions are impossible if they: add a property to a list which already has that property, add a relationship to the previous candidate list when there is no candidate list, or if they would allow the next action to not have a relationship to the previous candidate list while the candidate list is not empty.

## C.3 BioFuelQR Dataset

Our application focus is driven by the design of catalysis for reverse order gas reaction that is key to generation of synthetic biofuels with higher selectivity (Canakci and Van Gerpen, 1999; Daza and Kuhn, 2016; Kattel et al., 2017; Artz et al., 2018; Stolarczyk et al., 2018; Xu and Carter, 2018; Mukhtar et al., 2022).

Questions in the BioFuelQR dataset uses the following template:

`What are the top-3 {catalyst label} {candidate list statement} that perform the RWGS reaction at a lower temperature (<200 C) and demonstrate higher adsorption energy for both CO2 and H2 (or facilitates both CO2 and H2 adsorption)?. {include statement} {exclude statement} Provide scientific explanations and return a list of top 3 answers and their explanations as a list of pairs. Let's think step-by-step.`

Table 3: List of actions and their possibilities.

| Action Type | Possible Values | # possible |
|---|---|---|
| Add Property to Include | high activity, high selectivity, high stability, novelty, low cost, low toxicity, high surface area, high porosity, crystal facet, availability | 11 |
| Add Property to Exclude | low activity, low selectivity, low stability, high cost, high toxicity, low dispersion, low porosity, high scarcity | 9 |
| Change Catalyst Type | unary catalyst, binary catalyst, trinary catalyst, catalyst | 4 |
| Toggle Oxide | on/off | 1 |
| Change Relation to Prev. Answer | including elements that are different from, including elements similar to, introducing new elements to, including elements from | 4 |
| Repeat Prompt | N/A | 1 |

## C.4 Baseline implementations

Here we define the parameters for the evaluations of the Baseline and MCR methods.

**Chain-of-Thought (CoT)** For the CoT baseline, we generated a prompt for each query following the templates described in Appendix C.1. We evaluated 9 adsorbates from the Open Catalysis Dataset and 2 prompts from the BFR dataset. For CoT, we simply send one prompt to the LLM to generate a list of candidate catalysts, including the phrases "Provide a scientific explanation" and "Let's think step-by-step". The reward of the result is reported.

**CoT with Self-Consistency** For the self consistency baseline, the query was evaluated 10 times independently using the same prompt from CoT. We checked the answer for consistency. However, there was no consistency between the top-$k$ answers from the LLM over the 10 trials. Perhaps due to the large diversity in catalyst compositions. Thus, the reward estimate returned in Table 1 is simply the maximum reward over the 10 trials.

**Tree-of-Thoughts (ToT)** For ToT, keeping computational cost in mind, we set a branching factor $b = 6$. This controls the number of nodes expanded at each point in the search. Thus, at each level the nodes with the top 6 rewards are expanded. To re-

duce computational cost, we restricted the number of actions to the top 12 actions with the highest prior probability $p(P, a_i)$. This way, we reduce the number of actions simulated at each step. If there are not 12 actions with nonzero prior probability for a node, we generate as many children as possible. This happens, for instance, at the second level of the search tree, where the action "change relation to previous answer" must be taken (of which there are 4 possibilities). This is because they will pass their candidate catalysts to their successor prompts. The ToT method was run for 5 steps to generate a tree with depth 5. If all actions were possible at every level, we would generate 300 nodes in BFS (not including the root node), but only 252 nodes were generate on average. Still, we were able to select at least 6 best nodes at each level. We did not experience a similar discrepancy in MCR because MCR has a more flexible branching policy. The average observed number of nodes in the final trees is reported in Table 1.

We did not include the depth-first-search method from Tree-of-Thoughts because our search does not support a specific ending criterion.

**MCR** For MCR, we set a discount factor, $\gamma = 0.9$ and exploration-exploitation trade-off of $c = 15$ to control the branching and depth of the search tree. Generally, decreasing $\gamma$ decreases the length of chains in the search tree while increasing $c$ increases the branching of the tree. We generated 300 nodes after the root node, meaning 301 nodes were in the final search tree.

MCR utilizes the policy in Equation 2 to determine which actions to carry out at which step. However, the policy must be modified in two cases. First, if a node is a leaf node, the policy is replaced by the prior probability distribution over actions, $p(P_t, a_i)$ (see Section 2). Secondly, if a node action pair has no visits ($N(P_t, a_i) = 0$) then the first term of Equation 2 is dropped to avoid dividing by zero.

## C.5 Reward Query

To query the language model to return adsorption energies, we use another prompt template:

```
Generate a list of adsorption energies,
in eV, for the adsorbate {adsorbate} to
the surface of each of the following
catalysts: {candidate list}. Return the
adsorption energies as a list of only
{len(candidate list)} numbers in the order
```

specified.

The LLM should return a list of numbers which can be averaged to produce a final energy. Since adsorption energies are negative we take the absolute value of the numbers listed by the LLM. units are in eV. If multiple adsorbates are given, as in the BFR examples, multiple prompts are generated and the results are summed over. Occasionally, the LLM does not give an output that can be easily parsed into a list of floats. In these cases, the query is re-run a maximum of 3 times. Such examples include but are not limited to uncommon delimiters and sporadic phrases in the output.

## D   Qualitative Analysis

| Questions | Answers | Reasoning criteria |
|---|---|---|
| What are the top catalysts with higher adsorption energy for both $CO_2$ and $H_2$ (or facilitates both $CO_2$ and $H_2$ adsorption) | Noble metal catalysts such as Pt, Rh, Pd, Ru supported on reversible metal oxide i.e., CeO2 (cerium oxide), TiO2 (Titanium dioxide)

While noble metals are active for hydrogen adsorption, reversible metal oxide facilitates the CO2 adsorption. The oxygen vacancy present in the reversable metal oxide facilitates C-O bond cleavage of CO2. Generally, interface sites are coined as the active sites. Higher metal-support interaction is key for their high activity. | **Adsorption energy**

**Electronic structures**

**Metal-support interaction** |
| Identify the top catalysts that exhibit weak adsorption energy for CO (product) | Metal catalysts such as Au, Ag, Cu, Zn demonstrate weak adsorption energy corresponding to CO | **Adsorption energy** |
| What are the top catalysts that perform RWGS reaction at lower temperature (<200 °C) | Atomically dispersed Pt, Rh, Pd and Ru catalysts on CeO2 or TiO2. Atomically dispersed metal sites bind CO very weakly due to their unique electronic structure consequently exhibits high selectivity to CO | **Electronic structures** |
| Is RWGS reaction structure sensitive? | Yes. Particles that are typically <2 nm are more active for the RWGS reaction. smaller particle size enables higher metal-support interaction which facilitates the CO2 activation and facilitates the reaction. In general step and kink sites i.e., coordinatively unsaturated sites (such as 110 planes) exhibit weaker CO binding energy. | **Electronic structures**

**Adsorption on specific crystal planes** |
| What are the crystal planes that are most active for the adsorption of CO2 for noble metal catalysts? | Open structure such as (100) or (110) planes of metal catalyst are more active towards CO2 activation. Binding energy of CO2 is higher on these crystal surfaces | **Surface structure**

**Adsorption on specific crystal planes** |
| What are the top bimetallic catalysts that are cheap as well as demonstrate higher adsorption energy for both CO2 and H2 (or facilitates both CO2 and H2 adsorption) | Metal oxide supported PtRe, PtCo, PtNi bimetallic alloys demonstrate higher adsorption energy for both CO2 and H2. Addition of Re, and other transition metals reduces the cost of Pt based catalyst | **Structural properties** |
| Identify which three catalysts bind phenol during hydrodeoxygenation reaction with High Conversion. | Zeolite supported Pt, Rh and Ru catalyst. Pt, Rh and Ru binds phenol strongly and activate for the hydrogenation. Acidity of zeolites facilitates C-O bond cleavage i.e., hydrodeoxygenation | **Structural properties**

**Adsorption energy** |
| Identify which three catalysts can bind acetic acid in hydrogenation reaction with High Conversion. | Ru/C, Ru/Al2O3 and Pt/C. Ru/C catalysts. Ru/C exhibited higher conversion compared to other support materials | **Structural properties**

**Metal-support interaction** |

Figure 4: Example queries from the BioFuelQR dataset representing reasoning with different combinations of chemical descriptors.

Figure 5: Example question and human answer from our compiled QA-dataset.

Step 1: Identify referred catalysts for the RWGS reaction The reverse water-gas shift (RWGS) reaction is used to convert CO2 and H2O into CO and H2. It is an important reaction in the Haber-Bosch process for the manufacturing of ammonia. Many catalysts can facilitate the RWGS reaction, including, e.g., transition metal oxides, ionic liquids, or zeolite-based materials.

Step 2: Identify the catalysts' desired properties The desired properties for a catalyst to perform the RWGS reaction at low temperatures (<200°C) and to demonstrate higher adsorption energy for both CO2 and H2 include: high catalytic activity, high selectivity, good thermal stability, and robustness. Additionally, the catalyst should demonstrate good adsorption capability for both CO2 and H2.

Step 3: Identify the top-3 catalysts The top-3 catalysts for the RWGS reaction at a lower temperature (<200 C) and demonstrate higher adsorption energy for both CO2 and H2 include:

Transition metal oxides: Transition metal oxides (e.g., iron oxides and copper oxides) demonstrate high adsorption energy for both CO2 and H2, and provide high catalytic activity at low temperatures (<200 C). Additionally, they are chemically and thermally robust.
Ionic liquids: Ionic liquids can provide high selectivity and good thermal stability for the RWGS reaction. Due to their high surface area, they can also promote good adsorption capability for both CO2 and H2.
Zeolite-based materials: Zeolite-based materials provide good thermal stability and robustness for the RWGS reaction. Additionally, they exhibit higher adsorption energy for both CO2 and H2 when compared to other catalysts.

Figure 6: Response to above query returned by Chain-of-Thought promting with GPT-3.

1. Ruthenium Oxides; Ruthenium oxides have been studied as catalysts for a variety of reactions, including the RWGS reaction. These catalysts have very high binding energy and are relatively inexpensive, making them a good candidate for this purpose.
Explanation; Ruthenium oxides possess a high binding energy which enables them to achieve the RWGS reaction at very low temperatures (<200C) and demonstrate strong adsorption properties both for CO2 and H2, making them well-suited for this purpose.

2. Hafnium Oxides; Hafnium oxides have been studied for their ability to catalyze the RWGS reaction with high binding energy and low cost. These catalysts are very stable and have been found to have stronger adsorption for both CO2 and H2 compared to the Ni-Co, Zeolite-Supported Metal Oxides, and Doped Titanium Oxide catalysts.
Explanation; Hafnium oxides are a good choice for the RWGS reaction as they have higher binding energy, and can catalyze the reaction at lower temperatures (<200C) with higher adsorption rates for both CO2 and H2, as well as being cost effective.

3. Platinum-Ruthenium Alloy; Pt-Ru alloys have been studied for their potential use as catalysts in a variety of reactions, including the RWGS reaction. They are able to promote the reaction at low temperatures (<200C), and have been found to achieve a high binding energy and strong adsorption for both CO2 and H2 due to the combination of the two metals in the alloy.
Explanation; Platinum-ruthenium alloys have been found to be effective catalysts for the RWGS reaction at low temperatures (<200C) due to their high binding energy, and also due to their strong adsorption properties for both CO2 and H2. These alloys are also relatively inexpensive, making them a good choice for this purpose.

Figure 7: Response to above query returned by MCR.

**1) Quality: How did the AI methods matched your answer?**
**Answer:** One answer from GPT-3.5 can be considered partially correct (transition metal) while the Monte Carlo Reasoner partially matched my answers and reasoning for the noble metal catalysts' RWGS activity. Both models were not able to address the requirement of catalyst activity of less than <200C. The Monte Carlo Reasoner identified noble metals, Platinum and Ruthenium. Hafnium was something that I would not have considered. For catalysts that have high adsorption energies for $CO_2$ and $H_2$, my answers were three Pt-based catalysts ($PtRe/SiO_2$, $Pt/CeO_2$ and Na-doped $Pt/ZrO_2$). I also identified Ni-based (Ni/La-doped$CeO_2$, NiCu, Ni/Ce-Zr-O) and Cu-based ($4Cu-Al_2O_3$) catalysts from my research. My reasoning is that catalysts that would be expected to demonstrate higher adsorption energies for both $CO_2$ and $H_2$ would contain noble and base metals such as Pt, Ru and Ni supported on oxides with a high level of oxygen vacancies to facilitate high adsorption energies for both $CO_2$ and $H_2$. From the manuscripts that I reviewed that have tested RWGS at 200C, none resulted in any significant $CO_2$ conversion (>5%). Lastly, calculated equilibrium constants from another paper reported 0.0043 at 200C and 0.0830 at 400C.

**2) Specificity: Which AI method matched the specificity of your explanation?**
**Answer:** Both methods didn't completely match the specificity of my explanation, but I would choose the Monte Carlo Reasoner because it identified Pt, even as part of a bimetallic catalyst. However, even Pt catalysts do not have much activity (>5% $CO_2$ conversion) for RWGS at <200C.

**3) Reasoning: Which AI methods used similar chemical descriptors as yours to reason about?**
**Answer:** In part, the Monte Carlo Reasoner. It correctly identified strong adsorption properties for both $CO_2$ and $H_2$ for the noble metal catalysts.

**4) Did the AI method return any wrong answer?**
**Answer:** Yes, they both did. GPT-3.5's claim that the ionic liquid and zeolite were good catalysts for RWGS was incorrect. They were not identified as RWGS catalysts in my search. Transition metal catalysts, like Ni, Cu, and their alloys, were identified as potential RWGS catalysts but they are not active at <200C. The Monte Carlo Reasoner incorrectly identified Hafnium as a potential RWGS catalyst. However, I conducted a follow-on search because I am not very familiar with its chemistry. Hafnium seems to be able to activate $CO_2$ but whether it can produce CO selectively through RWGS was not conclusive.

**4) Are any of the AI-generated answers novel/superior to the human expert answer?**
**Answer:** The Hafnium suggestion was novel for me, but it was not superior to the human expert answer.

Figure 8: Comparison of MCR vs standard Chain-Of-Thought prompting (via GPT-3) by domain expert 1.

**1) Quality: How did the AI methods matched your answer?**
**Answer:** The answer provided by the Monte Carlo Reasoner is much closer to my expectation. Traditionally, the catalyst for the RWGS reaction is composed of both noble metal and reversable metal oxide, which activate H2 and CO2 respectively. The catalyst systems provided by the Monte Carlo Reasoner falls into this category and are expected to show activity towards RWGS reaction. However, their performance at lower temperature is still debatable and subjected to experiment. On the other hand, none of the catalyst provided by GPT-3.5 is known to be a good catalyst for H2 activation. Although their activity towards activation of CO2 has been demonstrated, performance of these catalysts at lower temperature is highly unlikely.

**2) Specificity: Which AI method matched the specificity of your explanation?**
**Answer:** The catalyst system provided by Monte Carlo Reasoner are more relevant to RWGS reaction. PtRu is known to activate both CO2 and H2 at lower temperature and thus, it would be an interesting system to consider for RWGS reaction at lower temperature.

**3) Reasoning: Which AI methods used similar chemical descriptors as yours to reason about?**
**Answer:** The catalyst system provided by Monte Carlo Reasoner considered adsorption energy of both CO2 and H2 as the descriptor. This is consistent as adsorption energy has always been used as major descriptor in heterogeneous catalysis. Consequently, the answers provided herein are associated with higher confidence.

**4) Did the AI method return any wrong answer?**
**Answer:** Zeolite materials and ionic liquid suggested by GPT-3.5 is not correct as these materials are not active for hydrogen activation and do not expect to show any activity towards RWGS. Although, transition metal oxide activate hydrogen but requires higher temperature. The catalyst systems provided by Monte Carlo reasoner are more relevant to RWGS reaction, however, the choice of the metal oxide catalyst system may not be correct as metal oxides are known to perform towards RWGS reaction only at higher temperature.

**5) Are any of the AI-generated answers novel/superior to the human expert answer?**
**Answer:** Suggestion for Hafnium oxide materials could be considered as novel as this has not been considered in the literature. but its activity towards RWGS reaction remains questionable.

Figure 9: Comparison of MCR vs standard Chain-Of-Thought prompting (via GPT-3) by domain expert 2.

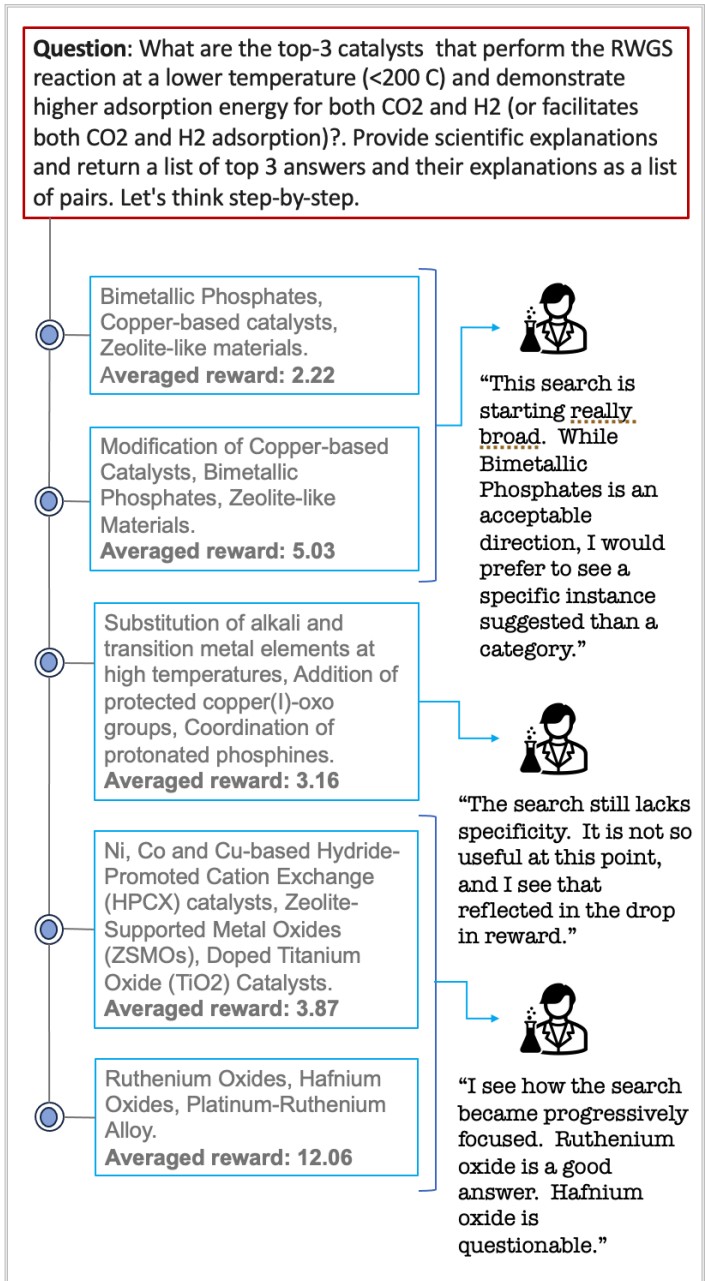

Figure 10: Illustration of an evaluation by a domain expert on the progression of top search results found on the path to the answer with highest reward.