# OpenReview forum: "Monte Carlo Thought Search: Large Language Model Querying for Complex Scientific Reasoning in Catalyst Design"
_EMNLP/2023/Conference — EMNLP 2023 Findings_

### Official Review · Reviewer_CnCD · 2023-08-01

**Soundness:** 5

**Excitement:**

4: Strong: This paper deepens the understanding of some phenomenon or lowers the barriers to an existing research direction.

**Paper Topic And Main Contributions:**

This paper targets at the scientific reasoning (i.e., catalysts discovering) problem by prompting LLMs. The authors propose a Monte Carlo Tree search method to retrieve the optimal prompt design, and a specially designed reward function is proposed. In experiment, they propose a new dataset and evaluate their method on it and a public dataset. Results show that their method can significantly outperform existing baselines on the catalysts discovering task.

The main contribution is: proposing an effective prompt searching algorithm as well as a reward function for the catalysts discovering task; introducing a new dataset.

**Reasons To Accept:**

Method: The proposed MCR method delicately reduces the search space for the prompt design, and it works good empirically.

Reward function: The proposed reward function is suitable for the specific chemical task.

**Reasons To Reject:**

No.

**Reproducibility:**

4: Could mostly reproduce the results, but there may be some variation because of sample variance or minor variations in their interpretation of the protocol or method.

**Reviewer Confidence:**

3: Pretty sure, but there's a chance I missed something. Although I have a good feel for this area in general, I did not carefully check the paper's details, e.g., the math, experimental design, or novelty.

---

> ### Author Rebuttal · Authors · 2023-08-28
>
> We sincerely thank the reviewer for their support of the work. We are
> passionate about demonstrating the impact of large-language models to
> complex scientific tasks. We specially sought to reverse engineer the
> mental reasoning processes employed by chemistry researchers as they go
> about designing experiments through long interviews and come up with a
> methodology that nearly codifies the process.
>
> We also recognize that even for organizations with nearly infinite
> resources, fine-tuning or engineering LLMs is destined to reach a point
> of diminishing returns, and it is more important to integrate
> well-established scientific concepts or functions than reinventing
> everything through language-modeling objectives.
>
> We sincerely hope our work will be recognized by the reviewers and
> community positively, and again thank you for your support.

---

### Official Review · Reviewer_etXD · 2023-08-02

**Soundness:** 4

**Excitement:**

4: Strong: This paper deepens the understanding of some phenomenon or lowers the barriers to an existing research direction.

**Paper Topic And Main Contributions:**

Authors propose a LLM-based model that solves a problem of suggesting the top-k catalysts for a given chemical reaction and generates the reasoning for each candidate. In particular, authors propose to use a Monte Carlo-based algorithm ("MCR") to prompt language model, and develop a domain-specific reward function. MCR algorithm solves the problem of searching over the large space of possible prompts that would result in the best outcome with respect to the reward function. Prompts are generated by recursively modifying a prompt node (starting with a root) with a set of available actions, creating tree-like structure. Then MCTS is applied to select the branches that show the most promising results. Finally, authors use a reward function that for each prompt queries the LLM to evaluate the adsorption energies, and use this feedback to guide generations. Additionally, authors build a new dataset for complex biological reasoning.

**Reasons To Accept:**

Authors propose an interesting approach for domain adaptation of the Large Language models. It is known that LLMs suffer from high sensitivity with respect to the input prompt: even small variation in the prompt can lead to significant differences in the generations. Authors propose to generate a wide variation of prompts by sampling from the action space and use MCTS with domain-specific reward to improve efficiency of the combinatorial search. The paper can be of interest to the wider research community seeking of ways to efficiently adapt latest SOTA LLM to narrow domains. Paper is well-written and contains relevant background information, detailed description and experiments backing up the proposed approach.

**Reasons To Reject:**

I don't have strong reasons to reject. As a suggestion, I would like to see more experimental results on different datasets.

**Reproducibility:**

4: Could mostly reproduce the results, but there may be some variation because of sample variance or minor variations in their interpretation of the protocol or method.

**Reviewer Confidence:**

3: Pretty sure, but there's a chance I missed something. Although I have a good feel for this area in general, I did not carefully check the paper's details, e.g., the math, experimental design, or novelty.

---

> ### Author Rebuttal · Authors · 2023-08-28
>
> 1.  **Reasons to reject**
>
>     1.  **I don\'t have strong reasons to reject. As a suggestion, I
>         would like to see more experimental results on different
>         datasets.**
>
> We sincerely thank the reviewer for their assessment of the work. Our
> submission provides initial empirical evidence to the merit of
> tree-based heuristic search approaches for LLM-based reasoning, and our
> further experiments confirm these findings. However, we fully agree with
> the reviewer that more experiments are needed to establish such findings
> on a rigorous basis. Such experiments should be the subject of future
> work. Our motivation to pursue a short paper was based on the promise of
> strong initial findings:
>
> 1.  The correlation between the performance of both our Monte Carlo
>     Reasoner and the contemporary Tree-of-Thoughts work is really
>     important. Even on a smaller scale study, both their outperformance
>     of other reasoning approaches such as Chain-of-Thought and
>     Self-Consistency etc. suggests that tree-based methods hold
>     significant promise for complex scientific domains such as
>     chemistry, or medicine.
>
> 2.  We were also astonished by the impact of a domain-specific reward
>     function. We believe integration of such reward functions can be key
>     to address model hallucination challenges for scientific domains.
>
> 3.  We also observed that the improved algorithmic performance also
>     comes with the price of significant inference cost and processing
>     latency, and that was another motivation to share our findings with
>     the community at the earliest opportunity. Parallelization of
>     tree-search methods is a well-studied topic in high-performance
>     computing literature, and we seek to draw the attention of the ML
>     systems research community to transition tree-based inference
>     methods like ours from promise to practice.

---

### Official Review · Reviewer_LJ9V · 2023-08-05

**Soundness:** 2

**Excitement:**

3: Ambivalent: It has merits (e.g., it reports state-of-the-art results, the idea is nice), but there are key weaknesses (e.g., it describes incremental work), and it can significantly benefit from another round of revision. However, I won't object to accepting it if my co-reviewers champion it.

**Paper Topic And Main Contributions:**

The authors apply Monte Carlo tree search (MCTS) to the problem of finding a query such that, when given as input to an LM, outputs recommendations for a chemical catalyst. They propose an iterative approach, where at each step, the current prompt is modified using a randomly selected action to produce a modified prompt, with the hope that the new prompt is better at performing the task.

**Questions For The Authors:**

A. What are the set of possible "mutations" of the prompt? Are they listed in Table 3 in Appendix C? This table is fairly difficult to understand.

 B. Can the same action be applied multiple times to a prompt? What would this look like?

 C. In Algorithm 1, on line 2, is the full tree initialized? Isn't the tree very large? What is P_t?

 D. In the comparison with ToT, why stop at depth 5? It seems like the search could have continued further, and the current comparison isn't as fair since ToT evaluates fewer prompts than MCR.

**Reasons To Accept:**

- The concept is interesting, with an interesting application of MCTS and LLMs to a real problem in chemistry research.

**Reasons To Reject:**

- The motivation in the Introduction is unclear. What is the search space? How is query decomposition helpful for searching in this space? Perhaps a concrete grounding example can help clarify.
 - It is difficult to understand the Monte Carlo Reasoner as described, and a figure containing an example would certainly help.
 - Experiments were run only on text-davinci-003, so they are not easily reproducible in the long-term. There were no experiments on open models, which could have alleviated the query cost, and enabled experiments on more queries from the dataset.
 - Many parts of the paper are unclear (see below) and difficult to follow.

**Reproducibility:**

4: Could mostly reproduce the results, but there may be some variation because of sample variance or minor variations in their interpretation of the protocol or method.

**Reviewer Confidence:**

4: Quite sure. I tried to check the important points carefully. It's unlikely, though conceivable, that I missed something that should affect my ratings.

**Typos Grammar Style And Presentation Improvements:**

Lines 65-67: I don't understand the meaning of R here. What would be an example of a possible prompt?

Lines 97-111: "reward function in LLM-based computations" How are reward functions used in LLM computations? Is the reward function for the MCTS?

Lines 136-139: It would greatly aid the reader's understanding if an example were provided of the original prompt and the new prompts after introducing "additional catalyst properties...such as 'composition of metals'". Further, what would a child node of this new prompt look like?

Line 143: From the perspective of the LM, isn't it the case that each prompt in the search tree is independent of the other prompts? If so, perhaps "reasoning pathway" is not a good term for this, as it suggests something akin to a chain-of-thought, where each step is a deduction of a new fact from premises given by previous steps. Since each node in the tree is not a reasoning step, I'm not sure "Monte Carlo Reasoner" is the most appropriate name either.

---

> ### Author Rebuttal · Authors · 2023-08-28
>
> 1.  **Reasons To Reject:**
>
>     1.  **The motivation in the Introduction is unclear. What is the
>         search space? How is query decomposition helpful for searching
>         in this space? Perhaps a concrete grounding example can help
>         clarify**
>
> Thank you for these thought-provoking questions. They are at the heart
> of our objective to apply an LLM-driven search for novel catalysts, and it is truly important that we
> are able to communicate the potential for impact and key scientific
> insights to the reviewers and any reader.
>
> > **"The motivation in the Introduction is unclear."**
>
> Let us begin with "motivation" with reference to Figure 1. As we mention
> in lines 28-36 (page 1), energy efficiency or low energy (or heat)
> requirements is a key consideration for discovering the next generation
> of chemical processes. Such consideration is at the core for converting
> waste to synthetic fuel that would be used to fuel airplanes, or
> producing steel which consumes an enormous amount of heat and
> contributes significantly to pollution.
>
> Mathematically, one could represent this as an optimization problem
> where we model a chemical process as a function and formulate the search
> problem as finding the optimal combination of all process parameters
> that minimizes a cost function modeled around energy efficiency.
>
> However, fields such as chemistry/catalysis research are highly
> empirical, where experts place a premium on their ability to reason
> based on scientific literature that dates back a century. This implies
> that the optimization process needs to be 1) human interpretable, and 2)
> the reasoning needs to follow key concepts from scientific literature.
> This opens the door to LLMs trained on scientific literature, and our
> work demonstrates how an automated search algorithm could mimic that
> mental search-and-elimination process employed by a human expert.
>
> > **"What is the search space?"**
>
> The search space is the collection of possible factors that can
> influence the design and performance of a catalyst. We interviewed
> catalyst researchers to codify their thinking processes and listed
> factors they themselves use in their mind to search for a catalyst.
> Table 3 in appendix C provides a complete list of these factors.
>
> "**How is query decomposition helpful for searching in this space?"**
>
> Our initial investigation of LLMs revealed that basic prompting (such as
> "what is a good catalyst for reaction X?") leads to basic answers that
> could be found on a wikipedia page. To quote our colleagues from
> chemistry, they do not like an answer "that feels read out of a science
> encyclopedia, not specific to any process or parameters". They prefer
> answers "that are grounded in domain specific concepts, like how they
> would explain a rationale to another colleague". In fact, several of our
> MCR guided answers were praised by them for their high specificity over
> the state of the art.
>
> Now, we employ two key ideas to achieve this high specificity. 1) We use
> the "adsorption energy" as a measure of reward, which is an important
> domain-specific measurement commonly used by catalyst researchers. 2) we
> "probe" the LLM with different combinations of prompt phrases.
> Identifying and ordering relevant phrases is critical in narrowing down
> the search for catalysts. For example, we may first prioritize "low
> cost", which directs the search towards a specific family of metal
> catalysts. Next, we may prompt the model to narrow the search based on,
> say, "crystal structures". These different considerations are visually
> demonstrated in Figure 1. The need for order and identification of
> relevant feature subspaces for construction of prompts and efficient
> probing of LLM's internal knowledge makes query decomposition a focal
> point of our research.
>
> "**Perhaps a concrete grounding example can help clarify"**
>
> In Figure 10 of the appendix, we provided concrete qualitative analysis
> of how the search works; we analyze how the LLM may use elements of the
> prompt to guide the selection of catalysts. With the extra page, we will
> move this figure into the main text and provide additional concrete
> details to clarify.
>
> 2.  **It is difficult to understand the Monte Carlo Reasoner as
>     described, and a figure containing an example would certainly
>     help.**
>
> We included a figure demonstrating the tree search and prompt mutations
> in figure 10 of the appendix. With the extra page, including this in the
> main text would clarify the confusions around the algorithmic design and
> the search space and query decomposition of the previous comment.
>
> 3.  **Experiments were run only on text-davinci-003, so they are not
>     easily reproducible in the long-term. There were no experiments on
>     open models, which could have alleviated the query cost, and enabled
>     experiments on more queries from the dataset.**
>
> We believe in the promise of our methodology, but as the reviewer
> pointed out, we also recognize the critical importance of
> reproducibility and increased experiments. We fully agree that using
> open models would have been highly desirable--in fact, we initially
> experimented with doing so. However, we finally decided to base our
> experiments on OpenAI models because:
>
> 1.  Our overall methodology strongly relies on the robustness of
>     LLM-instruction following. Robustness is critical as both
>     Tree-of-Thoughts and our methods pursue "deep chains" in the search
>     tree, where results from one internal node are transferred to the
>     searches spawned over its children. LLMs are an extremely
>     fast-moving field, and despite all the excitement around open source
>     models, we could not find any other model that offers both the
>     robustness of instruction following and coverage of scientific
>     literature. We had performed extensive experiments with Galactica
>     but it was not reliable for deep tree searches. We will include
>     these results in the final version.
>
> 2.  Preparing a robust open-source model with coverage of the scientific
>     literature would not be impossible, but it would require a
>     tremendous amount of resources. Any such investment is also risky in
>     a time when new models are released nearly every week.
>
> 3.  Our primary goal for this short paper was to demonstrate the
>     possible scientific reasoning capabilities and resulting impact of
>     LLMs. Since our goal is to demonstrate a new method to augment
>     scientific reasoning, we chose to focus on developing methodologies
>     for eliciting these scientific capabilities in lieu of LLM
>     engineering. Therefore, we chose the tool that helped to demonstrate
>     that point best.
>
> Besides the demonstration of scientific impact, we believe our work
> conveys other critical points for LLM research:
>
> 1.  Our methodology does not make any assumptions about a specific LLM,
>     and it would be trivial to swap a new LLM into our framework
>     (provided the LLM is sufficiently capable of instruction following).
>
> 2.  Our work demonstrates that tree-search methods have a strong value
>     proposition over existing methods for LLM reasoning (CoT,
>     self-consistency etc.). We would like to emphasize that
>     Tree-of-Thoughts (ToT) was released as a preprint in May and is
>     contemporary to our work--our methodology was developed separately.
>     Thus, an important contribution of our work is demonstrating the
>     merit of tree-based reasoning approaches for complex scientific
>     tasks. We note that ToT is primarily tested on toy datasets instead
>     of real-world, impactful problems--we show that tree-based
>     approaches are capable here. Related further experiments conducted
>     after submission strongly confirm the high performance of tree-based
>     search methods over existing approaches.
>
>     1.  **Many parts of the paper are unclear (see below) and difficult
>         to follow.**
>
> Thank you for elaborating on these points. Since our work is highly
> interdisciplinary, crafting a clear message is more challenging but of
> utmost importance. We have attempted to address your questions below and
> we will implement them into the final version to ensure it is clear and
> easy to follow.
>
> 2.  **Questions For The Authors**
>
>     1.  **What are the set of possible \"mutations\" of the prompt? Are
>         they listed in Table 3 in Appendix C? This table is fairly
>         difficult to understand.**
>
> We will clarify the technical details of the prompt generation in the
> final version. As an example, . Further, we will also include an example
> of a reasoning pathway to clarify what the actions do to the prompt.
>
> 2.  **Can the same action be applied multiple times to a prompt? What
>     would this look like?**
>
> The same action can be applied only if it does not cause a phrase to be
> repeated. For example, if the algorithm has already taken the action to
> consider catalysts with "low cost" in the prompt, it won't take the
> action to consider "low cost" again (we don't consider catalysts with
> "low cost and low cost"). We discussed this briefly in Appendix C.2
> (lines 863-871); we will further clarify this explanation along with the
> improvements for Table 3. We will include a detailed list of which
> specific actions can be taken multiple times.
>
> 3.  **In Algorithm 1, on line 2, is the full tree initialized? Isn\'t
>     the tree very large? What is P_t?**
>
> Initially, the tree in Algorithm 1 only contains a single node, the root
> node. As the algorithm runs, new nodes are added to the tree using the
> set of actions. For example, if the phrase "considering catalyst with
> low cost" is added to a prompt (a node), then a new child node is
> created containing the new prompt. Once a new node is created, the
> answer and reward are stored in the node. The number of nodes depends on
> the use case. For our tests, we used 301 nodes per tree, as shown in
> Table 1. P_t is the node selected by MCR in the tree T at depth t in
> each step. We interpret P_t as the t-th prompt in a reasoning pathway.
>
> 4.  **In the comparison with ToT, why stop at depth 5? It seems like the
>     search could have continued further, and the current comparison
>     isn\'t as fair since ToT evaluates fewer prompts than MCR.**
>
> We appreciate the Reviewer's comment. While the ToT method had fewer
> nodes, we still demonstrate an advantage. In Table 1, we included the
> maximum depth of each ToT and MCR. MCR was able to achieve a depth of
> \~9 in each evaluation with a reasonably few number of extra nodes over
> ToT. At each step in the ToT process, 6\*12=72 prompts are created. So,
> it would take 288 more steps to reach a depth of 9. We set the stopping
> depth at 5 to generate close to 300 nodes, which is the same number of
> nodes as MCR in this work. However, ToT was not able to generate the
> full 12 actions for every node at every level. This is because some
> actions cannot be taken multiple times and actions are ignored if the
> adsorption energies cannot be parsed out of a reward answer. This means
> that some steps may generate less than 6\*12 nodes.
>
> We again would like to note that ToT is contemporaneous work which was
> released only a month prior to our submission. Hence, our comparisons to
> it were not required, but we included them to further strengthen our
> overall message; tree-based reasoning can enable LLMs to perform well on
> complex and impactful scientific reasoning tasks. In additional tests
> that we have done after submission, we have modified the ToT
> implementation to traverse more nodes. We find that this increases
> performance closer to MCR. Regardless, our overall results show that
> tree-based models are quite effective for complex reasoning problems in
> the scientific domain, which was not shown by ToT.
>
> 3.  **Typos Grammar Style And Presentation Improvements:**
>
>     1.  **Lines 65-67: I don\'t understand the meaning of R here. What
>         would be an example of a possible prompt?**
>
> R is the optimal set of catalyst properties to reason over. Our goal
> with the tree search methods is to use a reward function to search for
> R. We will replace this sentence with a more thorough explanation of the
> goal of our prompt search.
>
> 2.  **Lines 97-111: \"reward function in LLM-based computations\" How
>     are reward functions used in LLM computations? Is the reward
>     function for the MCTS?**
>
> By "LLM-based computations", we meant using an LLM to perform scientific
> reasoning. It has been common to apply LLMs with CoT-based prompting to
> mathematical reasoning problems. We adopted this terminology from that
> area of research. MCR and ToT are just two types of LLM-based
> computation methods that use a reward function to guide which actions to
> apply to a prompt. The LLM-computation is the whole process of taking an
> initial query and returning an answer.
>
> 3.  **Lines 136-139: It would greatly aid the reader\'s understanding if
>     an example were provided of the original prompt and the new prompts
>     after introducing \"additional catalyst properties\...such as
>     \'composition of metals\'\". Further, what would a child node of
>     this new prompt look like?**
>
> We believe our planned inclusion of figure 10 from the appendix will
> help explain our prompt creation methodology.
>
> 4.  **Line 143: From the perspective of the LM, isn\'t it the case that
>     each prompt in the search tree is independent of the other prompts?
>     If so, perhaps \"reasoning pathway\" is not a good term for this, as
>     it suggests something akin to a chain-of-thought, where each step is
>     a deduction of a new fact from premises given by previous steps.
>     Since each node in the tree is not a reasoning step, I\'m not sure
>     \"Monte Carlo Reasoner\" is the most appropriate name either.**
>
> We had two reasons to call these pathways "reasoning pathways". 1) the
> sequence of actions taken in the pathways is determined by the reward at
> each step. Thus, the path from the root query to a leaf of the search
> tree is a sequence of decisions based on heuristic feedback. This is
> akin to how scientists would rationally deduce a set of catalysts and
> catalyst properties by moving from one inquiry to the next. 2) each step
> of the reasoning pathway uses catalysts from the previous prompt to
> condition the LLM to produce the next catalysts to be either "similar
> to", "different from", or "incorporating elements of" the previous
> catalysts. So, the reasoning pathways we define are not independent
> prompts--the LLM is reasoning based on intermediate results just like
> CoT. Figure 10 in the appendix, which we will include in the final
> version of the paper, shows how the catalyst properties and catalysts
> are passed between prompts in a reasoning pathway.
>
> The strength of our Monte Carlo Reasoner (MCR) method and the
> contemporaneous ToT method is their ability to search branching
> reasoning pathways. This is the key distinction from CoT; MCR uses a
> reward-function to select the best path in the tree to follow. The MCR
> method uses random sampling (MCTS) to determine which branches to use at
> each step, under the assumption that the highest reward branch may not
> always lead to the best catalysts. Since the MCR combines Monte Carlo
> and reasoning pathways, we decided to call it the Monte Carlo Reasoner.

---

### Meta-Review · Area_Chair_T3sH · 2023-09-19

**Recommendation:** 4

**Metareview:**

The authors proposed an approach for domain adaptation of the Large Language models.
The 2 out of 3 reviewers selected highly positive score (one selected "borderline")

---

### Decision · Program_Chairs · 2023-10-07

**Decision:**

Accept-Findings

**Comment:**

The authors proposed an approach for domain adaptation of the Large Language models.
The 2 out of 3 reviewers selected highly positive score (one selected "borderline")